# Microemulsions: An Encapsulation Strategy to Increase the Thermal Stability of D-limonene

**DOI:** 10.3390/pharmaceutics15112564

**Published:** 2023-11-01

**Authors:** Bruna Rodrigues Belem, Gustavo Vaiano Carapeto, Michele Georges Issa, Humberto Gomes Ferraz

**Affiliations:** Department of Pharmacy, School of Pharmaceutical Sciences, University of Sao Paulo, Professor Lineu Prestes Avenue, Sao Paulo 05508-580, Brazil; gustavo.carapeto@alumni.usp.br (G.V.C.); michelegeorges@usp.br (M.G.I.)

**Keywords:** D-limonene, microemulsions, oxidative stability, thermal stability

## Abstract

D-limonene, derived from citrus essential oils, holds significant therapeutic potential but faces challenges due to its high volatility, especially in pharmaceutical formulations. This study investigates microemulsions as a promising delivery system for volatile compounds, emphasizing their thermal protection for D-limonene. The formulation development was guided by a pseudo-ternary phase diagram and involved assays with different surfactants. Microemulsions were achieved solely with Labrasol^®^ (Gattefossé Brasil, São Paulo, Brazil), encompassing concentrations of 7.1% to 30.8% D-limonene, 28.6% to 57.1% Labrasol^®^, and 20.0% to 64.3% water. All formulations were homogeneous, transparent, and presented low viscosity, with adequate D-limonene content, indicating that the production is feasible at room temperature. While the formulations demonstrated robust physical stability under mechanical stress, they exhibited destabilization at temperatures exceeding 50 °C. In terms of oxidative stability, pure D-limonene exhibited an induction period of 4.88 min, whereas microemulsions extended this period by four to eight times. Notably, the induction period of the microemulsions remained practically unchanged pre and post-heating (70 °C), suggesting the formulation’s ability to enhance the D-limonene thermal stability. This highlights the value of oxidative stability analysis as a quicker tool than conventional oxidative tests, while affirming microemulsions as a viable encapsulation strategy for D-limonene protection against elevated temperatures.

## 1. Introduction

D-limonene is a byproduct of the citrus essential oils’ beneficiation process, widely employed as a flavor and fragrance additive in perfumes, soaps, and food products. From a pharmacotherapeutic perspective, it is also a substance of great interest due to its therapeutic activities, including relief of heartburn, improvement of peristalsis, and treatment of gastroesophageal reflux [1,2].

However, the availability of D-limonene as an active substance for the population is limited and challenging, since it is a highly volatile component, whichhinders its incorporation into pharmaceutical forms. In this context, the development of innovative formulations capable of incorporating volatile substances becomes a matter of interest. In recent years, microemulsions have received attention as one of the most promising drug carrier systems due to their high system stability, ease of preparation, and ability to increase the active substance stability [3,4].

These systems have also been shown to be effective in encapsulating volatile substances, increasing the shelf life and processability of essential oils [3,4]. However, many of the studies conducted so far report partial or incomplete data, highlighting the importance of new works that expand the exploration of the entire technical capacity of microemulsions, ensuring that formulations containing essential oils have the adequate quality and safety to be made available to the population [4,5].

As an illustrative instance, studies by Karadağ et al. (2021) [6], Toledo et al. (2020) [7], and Laothaweerungsawat et al. (2020) [8] adeptly demonstrate the capacity of microemulsions to potentiate the therapeutic effects of essential oils. In the case of *Rosmarinus officinalis* L. oil, the microemulsion effectively enhanced its antimicrobial activity [6]. Microemulsions of *Cymbopogon nardus* exhibited improved in vitro and in vivo anti-Candida albicans activity [7], while microemulsions of *Origanum vulgare* L. facilitated enhanced transdermal delivery of carvacrol [8]. However, scant information has been provided regarding the stability of such compositions, particularly concerning oxidative stability, despite the utilization of gas chromatography coupled with mass spectrometry (GC-MS) or high-performance liquid chromatography (HPLC) to acquire crucial insights into the composition of the utilized essential oils.

Nonetheless, in numerous instances, an additional impediment to the development of essential oil products lies in the challenge of quantifying these substances, often necessitating the utilization of robust and costly techniques, such as GC-MS [9]. In this context, the search for more accessible alternatives, such as oxidative stability, is ongoing. This technique, in addition to being innovative, is more economical, quick, and highly sensitive to the oxidation reactions of essential oils. Consequently, it can serve as a feasible alternative for indirectly assessing the content of volatile compounds in pharmaceutical products, as well as characterizing microemulsions in terms of their ability to protect and/or delay the volatilization of essential oils [10].

Although it is possible to find in the literature studies such as those of Mehanna, Abla, and Elmaradny (2020) [11], who also investigated the development of D-limonene microemulsions, concluding that microemulsion is a promising tool for reducing the volatility, improving stability, and masking dermal irritants of essential oils, thermal and oxidative stability studies were not conducted, which are important characteristics that need to be additionally assessed for better comprehension of these formulations, providing new insights for the development of such products.

Therefore, the aim of this study is to evaluate the ability of microemulsified systems to protect D-limonene against volatilization, as well as to obtain formulations capable of incorporating essential oils, ensuring their stability, appropriate administration route, and dose control, applying the oxidative stability technique as an innovative tool of characterization.

## 2. Materials and Methods

### 2.1. Materials

The materials described in Table 1 were used for the development of D-limonene microemulsion formulations.

### 2.2. Methods

#### 2.2.1. Statistical Design

The development of microemulsions was planned by constructing a pseudo-ternary phase diagram. Initially, 10 formulations (Table 2) contained in the pseudo-ternary phase diagram (Figure 1) were selected for production, to cover all regions of the triangle and limit the transition areas between systems through the macroscopic analysis of each one.

#### 2.2.2. Definition of Surfactants

To determine the appropriate proportions of each surfactant for the formulation, the Equations (1) and (2) presented below were employed [12].
A + B = 1(1)
(A × EHL_A_) + (B × EHL_B_) = EHL_req._(2)
where:

A = surfactant A;

B = surfactant B;

EHL_A_ = HLB value of surfactant A;

EHL_B_ = HLB value of surfactant B;

EHL_req._ = required HLB value in the mixture.

In order to determine the appropriate excipient for the formulation, a comprehensive evaluation was conducted involving a range of materials (Table 3), either individually or in binary combinations. The surfactant concentrations provided in Table 2 corresponded to the constituents of each binary combination and/or single surfactant as delineated in Table 4. In other words, each combination was utilized in the production of the 10 formulations outlined in Table 2, resulting in a total of 130 distinct formulations. This expansive array was established to delineate the microemulsion region associated with each surfactant, thereby facilitating the identification of the most suitable excipient, along with its optimal concentration.

The selection of surfactants was made based on the required HLB of D-limonene [15]. According to Kourniatis et al. (2010) [16], orange terpenes have an HLB value of 6.4, while orange oils have an HLB value of 8.7. However, during preliminary studies, promising formulations were obtained in the HLB value of 12. Thus, the following mixtures (Table 4) were prepared to obtain the surfactant phase with HLB values of 6.4, 8.7, and 12.

#### 2.2.3. Formulation Production

Initially, binary mixtures of surfactants listed in Table 4 were prepared, and subsequently, the surfactant, water, and D-limonene were weighed sequentially into a test tube in sufficient quantity to yield 10 mL of the formulation. Then, manual agitation was carried out, and the macroscopic characteristics of the system, such as transparency, homogeneity, and low viscosity, were observed to ascertain the spontaneous formation of microemulsion. The entire process was conducted at room temperature.

The optimal proportions of aqueous phase, oil phase, and surfactant facilitate the spontaneous formation of microemulsion without requiring significant mechanical and/or thermal energies for production. Therefore, while preparing the formulations as described in Table 2, priority was given to the ones that could undergo spontaneous formation, with the aim of optimizing and simplifying the process.

##### Titration Method

Once the formulations with potential for microemulsion formation were identified, a region of the ternary phase diagram was delimited and explored by the titration method. This method involves the preparation of a mixture of surfactant and oil phase, with known proportions, followed by the incremental addition of small quantities of water, mimicking an analytical titration. Following each water addition, a simple agitation was performed, and subsequently, the resulting product was macroscopically characterized to ascertain the presence or absence of microemulsion formation [17]. To explore the microemulsion areas, eight different surfactant and D-limonene proportions were used (Table 5), starting from 100 mL of each mixture for better system visualization within a Schott flask.

Given the precise measurement of the added water, it is feasible to calculate the proportions of the three components of the formulation at each step, elucidating the pseudo-ternary phase diagram with respect to the microemulsion regions.

#### 2.2.4. Microemulsions Characterization

##### D-limonene Content

The assessment of the D-limonene content in the microemulsions was carried out according to the UV-Vis spectrophotometric analytical method developed by our research group [18]. For this purpose, the samples were diluted in absolute ethanol as a solvent system. Subsequently, the microemulsions were quantified using an Evolution 201 spectrophotometer (Thermo Fisher Scientific, Walthman, MA, USA) at 280 nm. This wavelength was selected due to the absence of signal interference from Labrasol^®^ at its highest concentration in formulation, as well as the absence of a signal from the absolute ethanol as the solvent system (Appendix A). The measurements were executed with a 5 mm optical path length, through an analytical calibration curve.

##### Macroscopic Analysis

To evaluate the stability and physical characteristics of the formulations, the microemulsions were left to rest for at least 24 h after preparation. At the end of this period, the formulations were observed and characterized by color, homogeneity, phase separation, and presence of precipitates.

##### pH

The pH of the formulations was measured in triplicate using a digital pH meter PG 2000 (Gehaka, São Paulo, Brazil) equipped with a pH/ORP electrode DME-CV8 (Digimed, São Paulo, Brazil), which was previously calibrated with Ph 4.0 and Ph 7.0 buffer solutions at room temperature.

##### Density

The microemulsion density was assessed using the portable density meter DMA 35 (Anton Paar, Graz, Austria), whose operating principle relies on the application of the U-tube oscillation method. Initially, a verification of ultrapure water at 20 °C (Water Check) was performed, and upon obtaining consistent results, the evaluation of the sample proceeded.

##### Centrifugation

After a minimum resting period of 24 h, the physical stability of the microemulsions was evaluated by mechanical stress through centrifugation. The samples were subjected to a rotation of 4000 rpm in cycles of 15, 30, and 60 min using the Rotofix 32 centrifuge (Hettich, Kirchlengern, Germany), using a methodology adapted from Carnicel (2014) [19]. After these periods, the samples were again evaluated macroscopically and classified as stable, transparent, turbid, or with phase separation.

##### Oxidative Stability

To evaluate the capacity of microemulsions in protecting D-limonene against volatilization, the accelerated oxidative stability technique was employed using the RapidOxy 100 (Anton Paar, Graz, Austria). Utilizing a sample mass of 5 g, the method was set up with an initial temperature of 20 °C and a target temperature of 120 °C, along with a pressure of 400 kPa of pure oxygen in the analysis chamber. The purge of volatile samples was used to assess pure D-limonene, placebo formulations, and microemulsions containing D-limonene. The endpoint for the test was defined as a 10% decrease in pressure, which corresponded to the consumption of oxygen by the material and, thus, its oxidation.

##### Thermal Stress

The microemulsion formulations underwent thermal stress, in order to evaluate the ability of the product to resist pharmaceutical industrial processing, through a controlled heating process from 30 °C to 70 °C, with a heating rate of 5 °C per 30 min using a Carousel 12 Plus Reaction Station reflux system (Radleys, Saffron Walden, UK) in a total of nine steps. Following each heating step, a macroscopic evaluation was carried out to assess the formulations’ physical stability, categorizing them as either stable, transparent, turbid, or exhibiting phase separation. After the last cycle of heating (70 °C), the microemulsions were evaluated once again regarding the oxidative stability, pH, and density.

##### Multivariate Analysis of Microemulsions

In order to do a comparison of microemulsions formulations, a multivariate analysis of principal components (PCA) using the Statistica^®^ software version 14.0.0.15 (TIBCO^®^ Software Inc., Palo Alto, CA, USA) was performed, considering the data from characterization before and after the thermal stress. After the data were standardized, the bidimensional plot was constructed, considering the two principal components (PC) with the highest eingenvalues.

#### 2.2.5. Hydrophilic–Lipophilic Balance Determination

Given the scarcity of data regarding the HLB value of D-limonene in the literature and the results obtained in Section 2.2.2 of this study, experiments were conducted to determine the HLB value of D-limonene. To accomplish this, a method based on Ferreira et al. (2010) [20] was used, consisting of testing the formation of stable emulsions on a defined HLB range. Drawing from the outcomes of microemulsion formation, the HLB range of 9 to 15 was investigated by formulating emulsions containing 10% D-limonene, 85% ultra purified water, and 5% surfactant system. This system included either Span 85 or Labrasol^®^ combined with Tween 80, in varying proportions, in order to achieve the required HLB value (Table 6), also calculated by Equations (1) and (2).

Subsequently, to verify the emulsion’s physical stability, the dispersion analyzer LUMiSizer equipment (LUM, Berlin, Germany) was used to obtain each emulsion’s instability index. For the emulsion analysis, aliquots were transferred to a 2 mm path length polyamide cuvette. The samples underwent a 4000 rpm centrifugation at 25 °C using the dispersion analyzer LUMiSizer equipment (LUM, Berlin, Germany). A total of 800 profiles were obtained over a 5 s interval, employing a wavelength of 865 nm, totalizing 66 min for the analysis. Finally, the SEPView^®^ 6 software Version No 1.1.10 integrated with the equipment was used to perform a stability analysis based on the instability index.

## 3. Results

### 3.1. Microemulsion Development

#### 3.1.1. Delimitation of Pseudoternary Phase Diagram for Each Surfactant

The results obtained from Appendix A indicate that the mixtures of various surfactants, at appropriate proportions to achieve a required HLB of 6.4, failed to generate spontaneously forming microemulsion systems. It is plausible that the application of external energies such as elevated temperatures or shear forces might promote the formation of microemulsions in some of the compositions. Nonetheless, to ensure greater system stability and reduced the complexity of the production process, it is preferable that microemulsions form spontaneously. Thus, it can be concluded that the HLB value of 6.4 and/or the surfactants used to achieve this condition are unsuitable for D-limonene.

Subsequently, it was observed that only a formulation (Figure 2) composed of 20% aqueous phase, 10% D-limonene, and 70% surfactant (67.65% Labrasol^®^ with 32.35% Span 85) was capable of spontaneously forming a microemulsion considering an HLB requirement of 8.7, while the other compositions exhibited turbidity, phase separation, or resulted in emulsions and lotions (Appendix A). Moreover, it was also evident that after 24 h of preparation, formulation 8 remained stable, while formulations 17, 19, 30, 32, and 34 exhibited phase separation, with at least one of them being transparent, homogeneous, and low viscosity, which are well-established characteristics of microemulsions.

Afterward, the formulations obtained using a surfactant mixture to attain an HLB requirement of 12 (Appendix A) were evaluated. It was observed that Labrasol^®^ as the sole surfactant (Figure 3) was able to generate microemulsion systems, as demonstrated by formulations 17 (40% aqueous phase, 10% D-limonene, and 50% surfactant) and 19 (20% aqueous phase, 30% D-limonene, and 50% surfactant). Therefore, it is reasonable to conclude that the use of Labrasol^®^ is likely associated with the formation of microemulsions since, among all the evaluated combinations, three microemulsions were achieved using this surfactant, which was rationally chosen based on the definition of an HLB requirement.

Labrasol^®^, categorized as caprylocaproyl polyoxylglycerides, represents a non-ionic surfactant with primary utility as a solubilizing agent, a surfactant for microemulsions, and a lightweight foam generator when used in conjunction with pump devices, obviating the necessity for propellants. Its molecular composition encompasses a minor proportion of mono-, di-, and triglycerides, with a predominant presence of PEG-8 (MW 400), along with mono- and diesters derived from caprylic (C8) and capric (C10) acids [14,21]. The distinctive structural configuration of Labrasol^®^, divergent from the other evaluated surfactants, is presumably instrumental in the achievement of microemulsions. This is substantiated by the inability of Span 85-NV-LQ, Span 80 Pharma-LQ, Span 40 MBAL-PW, and Tween 60 NF MBAL-LQ to facilitate microemulsion formation, even at elevated concentrations.

Furthermore, formulation F8 (Figure 3) exhibited a slight phase separation, with one of the phases showing characteristics resembling those of a spontaneously formed microemulsion. Similarly, formulation F54 also presents a phase that appears transparent, free of visible particles and low viscosity, indicating that neighboring formulations in the pseudoternary phase diagram could potentially yield spontaneously formed microemulsions. Based on the findings in Figure 3, a ternary phase diagram (Figure 4) was constructed to identify microemulsion regions, as well as regions that will be investigated by the titration method.

Despite the fact that formulations with over 50% surfactant can lead to micellar systems, it is well-established that microemulsions typically form spontaneously with high concentrations of surfactants, sometimes necessitating the inclusion of a co-surfactant [3,4]. Moreover, it is widely recognized that the curvature, and consequently, the sizes of micelles formed in a micellar system differ from those in microemulsions. Micelles in microemulsions are typically smaller, resulting in a heightened solubilizing capacity for the system [22].

This holds particular significance for D-limonene formulations, as it is advantageous for the formulation to possess the capability to incorporate the highest possible concentration of the essential oil, especially considering its commonly elevated therapeutic dosages (25 to 300 mg/kg of body weight) [1,23,24,25]. Consequently, the approach to developing microemulsions is more compelling because it theoretically allows for a higher concentration of D-limonene. This, in turn, facilitates the administration of a smaller final product volume, presenting an intriguing strategy for this component.

Therefore, based on the findings so far, it can be inferred that Labrasol^®^ plays a crucial role in achieving microemulsions, as it was responsible for the formation of a microemulsion when used in its entirety or in combination with Span 85. Given the greater ease of the process and formulation, Labrasol^®^ was chosen as the sole surfactant for the formulations.

#### 3.1.2. Titration Method

At the end of each addition of water in the titration method, the formation or absence of microemulsions was observed. All the microemulsions obtained are depicted in Figure 5. Appendix A depicts the characteristics of the formulations obtained after the addition of 10 mL of water, followed by homogenization, in the first titration process. Image A1 represents a mixture of 55.0% D-limonene with 45.0% Labrasol^®^, and despite being a transparent, homogeneous, and low viscosity solution, it cannot be considered a microemulsion since it lacks an aqueous phase in its composition. Similarly, the other formulations (A2–A20) did not yield spontaneously formed microemulsions, as they all exhibit a milky appearance. Consequently, none of these compositions were considered for the continuation of this study.

Appendix A presents the characteristics of the formulations obtained after the addition of 10 mL of water in the second titration, followed by homogenization. Image B1 represents a mixture of 40.0% D-limonene with 60.0% Labrasol^®^, which does not meet the criteria for a microemulsion. Notably, formulation B4 (30.8% D-limonene, 46.2% Labrasol^®^, and 23.1% water) resulted in a spontaneously formed microemulsion, as the simple mixing of the three components provided a homogeneous, transparent, and low viscosity solution, as illustrated in Figure 5. The remaining compositions (B2, B3, B5–B20) were not considered for further study due to their milky appearance.

Appendix A illustrates the characteristics of the formulations obtained after adding 10 mL of water in the third titration process, followed by homogenization. Image C1 represents a mixture of 30.0% D-limonene with 70.0% Labrasol^®^, which also cannot be considered a microemulsion. It was observed that formulations C4 (23.1% D-limonene, 53.8% Labrasol^®^, and 23.1% water), C5 (21.4% D-limonene, 50.0% Labrasol^®^, and 28.6% water), C6 (20.0% D-limonene, 46.7% Labrasol^®^, and 33.3% water), and C7 (18.8% D-limonene, 43.8% Labrasol^®^, and 37.5% water) resulted in spontaneously formed microemulsions, as the simple mixing of the three components was able to provide a homogeneous, transparent, and low viscosity solution. The remaining compositions were not considered for the continuation of this study.

Appendix A depicts the characteristics of the formulations obtained after the addition of 10 mL of water in the fourth titration, followed by homogenization. Image D1 represents a mixture of 20.0% D-limonene with 80.0% Labrasol^®^, which also cannot be considered a microemulsion. It can be observed that formulations D5, D6, D7, D8, D9, D10, D11, D12, D13, D14, D15, D16, D17, D18, and D19, whose concentrations are described in Table 7, resulted in spontaneously formed microemulsions, as the simple mixing of the three components was able to provide a homogeneous, transparent, and low viscosity solution.

During the production process, it was also observed that formulations D5-D15 resulted in a microemulsion in less than 10 s of agitation, while compositions D16-D19 required 60 to 90 s of agitation. Therefore, it can be inferred that these latter compositions required greater mechanical energy for the formation of a microemulsified system and, therefore, are at the limit of the thermodynamic equilibrium condition within the ternary phase diagram. This is also confirmed by the fact that composition D20 was not able to spontaneously form a microemulsion, allowing us to conclude that the entire microemulsion region of the diagram has been explored.

Figure 6 illustrates the formulations obtained in titrations E, G, H, and I, respectively, providing a detailed view of the F54 region. This region was considered interesting since it contained formulations with higher concentrations of D-limonene. However, as can be seen in the images below, none of the compositions resulted in spontaneously formed microemulsions, so some formulations appeared cloudy and/or milky, while others showed phase separation. Therefore, none of the formulations resulting from titrations E, G, H, and I were considered for further study.

Based on what has been presented so far, it is possible to conclude that the microemulsion region of the ternary phase diagram is contained within the concentrations of 7.1 to 30.8% of D-limonene (oil phase), 28.6 to 57.1% of Labrasol^®^ (surfactant), and 20.0 to 64.3% of water, as represented in Figure 7 and Table 7.

It is worth highlighting, as demonstrated in Table 7, that while microemulsions commonly self-assemble and frequently require high concentrations of surfactants and co-surfactants to significantly reduce the interfacial tension between the oil and aqueous phases [3,4], it was possible to formulate a product with desirable properties using only approximately 30% of a single surfactant. In this particular scenario, the bending elastic energy, typically of secondary importance in relation to surface energy, gains greater relevance within microemulsion systems due to their remarkably low interfacial tension [26].

This is because the bending elastic energy is directly related to the shape of the formed micelle, with the droplet shape being the most common [26]. Given that these are oil-in-water compositions, it is understood that the D-limonene droplets are surrounded by a surfactant layer and dispersed in water within the obtained microemulsions. The surfactant layer, in turn, has a maximum limit of expansion. When this limit is reached, it leads to separation into various phase types, such as excess oil or water and presence of emulsions, among others. This phenomenon was likely observed during titration D, at the point where the D20 formulation was unable to reach thermodynamic equilibrium.

As the next step, a physicochemical characterization was conducted on select formulations obtained through titration, with the D-limonene content in each composition serving as the criterion for selection. In this context, formulations D12, D13, D14, D15, D16, D17, D18, and D19 were not characterized.

#### 3.1.3. Microemulsions Characterization

##### D-limonene Content, pH, and Density Characterization

The content of D-limonene was calculated using the linear equation (y = 0.0832x + 0.0062) (Appendix A) obtained by the authors of this study, and the results are shown in Table 8.

In order to obtain additional comparative parameters of similarity between the formulations, the microemulsions were also characterized in terms of pH and density. The pH values obtained for each of the microemulsions, in triplicate, are also presented in Table 8. It can be observed that all formulations have an acidic character, consistent with the nature of D-limonene, originating from citrus fruits.

Regarding the density values, it is observed that the lowest obtained value was equal to 0.9844 g/cm^3^, while the highest was 1.0199 g/cm^3^. In general, it can be stated that the obtained microemulsions are highly similar to each other in terms of density, as the deviation between them is approximately 0.01 (Table 8).

##### Macroscopic Analysis

As shown in Figure 8, all the formulations obtained are homogeneous, transparent, low viscosity, and without visible particles, consistent with the physical characteristics of a microemulsion. In addition, after 24 h of preparation, the thermodynamic equilibrium between the components of each formulation was not disturbed, indicating high physical stability for these compositions.

##### Centrifugation

The test aims to evaluate the physical stability of the microemulsions obtained under mechanical stress. After 60 min of testing, only formulation C4 showed a slight turbidity, as can be seen in Figure 9. This may indicate that the composition of 23.1% D-limonene, 53.8% Labrasol^®^, and 23.1% water is at the limit of its thermodynamic equilibrium, such that external disturbances can result in the rupture of the microemulsified system. However, after 5 min at rest, C4 returned to its transparent, homogeneous, and low viscosity state. Moreover, all other formulations remained stable throughout the assay, without apparent phase separation or turbidity.

Based on these findings, it is possible to infer that the microemulsions exhibit good mechanical stability, such that the applied stressful conditions of agitation and vibrations were unable to disrupt the system. Even the formulation C4 that showed higher sensitivity to agitation returned to thermodynamic equilibrium a few minutes after the cessation of agitation. Therefore, it is highly likely that these products are capable of withstanding normal conditions of transportation and handling, for example.

##### Oxidative Stability

The oxidative stability is evaluated by RapidOxy 100 through the maintenance of a constant temperature, while pressure variations during the assay are measured, resulting in the so-called induction period. The induction period is understood as the time elapsed between the beginning of the heating procedure until the moment when the formation of oxidation products increases rapidly, indicated by the pressure drop established in the assay termination criterion [10].

Thus, the longer the sample’s induction period, the more stable it is. In this context, it is possible to indirectly evaluate the microemulsion’s ability to protect the essential oil against volatilization by comparing the induction period of pure D-limonene, placebo formulation, and microemulsion containing D-limonene. The oxygen consumption was calculated by the equipment itself based on the difference between the maximum pressure reached during the assay and the pressure obtained at the end of the experiment.

It is observed that the maximum pressure obtained for each of the samples is higher than the programmed filling pressure (400 kPa). This is due to what is established by the Clapeyron equation (P × V = n × R × T), in which the pressure generated by the gas (P) multiplied by the volume occupied by it (V) is directly proportional to the number of moles (n) multiplied by the general gas constant (R) and the temperature to which the gas is subjected (T) [27].

In this context, it is understood that the increase in the assay temperature (20 °C to 120 °C) is responsible for causing the increase in pressure from 400 kPa to 513.5 kPa (Appendix A), as in the case of pure D-limonene. After the pressure and temperature stabilization, the sample starts consuming oxygen and, consequently, oxidation, resulting in the observed pressure drop.

The results obtained are expressed in Table 9 and show that all proposed microemulsions have been able to increase D-limonene stability. It is noted that this component in its pure form had an induction period of 4.88 min for 10.0% of oxygen consumption, while when incorporated into microemulsions, the period increased from 20 to 40 min for the same 10.0% consumption. Furthermore, it is observed that all placebo formulations presented an induction period superior to that of the proper formulation.

Figure 10 provides an example of the observed difference between the formulation and its corresponding placebo, indicating that Labrasol^®^ oxidation in water starts only after the oxygen consumption of the microemulsion is completed. Thus, it is understood that oxidation observed in microemulsions is mostly, if not exclusively, due to D-limonene. Although it may not be possible to identify D-limonene byproducts such as limonene oxide, carvone, and perillyl alcohol through this technique [28], it can serve as an alternative and indirect means to assess both the oxidation and volatilization of D-limonene in the absence of more robust techniques like GC-MS.

According to the oxidative results, it is also observed that formulations with a lower D-limonene content (D5 to D11) showed higher oxidative stability, considering that the lower concentration of the volatile oil makes it less available for oxidation reaction. However, even formulations with higher D-limonene content have been effective in protecting it within the microemulsion.

As illustrated in Figure 11, it is discernible that the microemulsion regions (B, C, and D) exhibit disparities primarily regarding the D-limonene content within the composition of each formulation, whereas formulation D demonstrates the lowest concentrations of this volatile component. It is worth mentioning that the most fitted model for those results was a quadratic function (R² = 0.9666). Therefore, it is notable that the influence of D-limonene content on the induction period is more pronounced at lower concentrations (titration B) compared to higher concentrations (titrations C and D). This implies that by reducing the D-limonene content by 50% in the formulation, there is an approximate 1.5-fold increase in the induction period. It is also interesting to note, as indicated by the results presented in Table 8, that the longest induction periods were achieved for the lowest concentration of D-limonene. This leads us to infer that its quantity within the composition is directly correlated with the microemulsion’s protective efficacy.

Upon closer examination of Titration C (Figure 12), it is discernible that neither the quantity of D-limonene nor that of water could disturb the thermodynamic equilibrium of the microemulsions. This is evidenced by the closely similar induction periods obtained for each, yielding approximately 22 min for the consumption of 10% of oxygen by D-limonene in the C formulations.

Titration D (Figure 13), on the other hand, unveils a robust linear correlation between the induction period and the amount of water present in each formulation (R² = 0.9560). It is notable that higher concentrations led to extended induction periods and consequently, enhanced stability of the formulations. This fact may be attributed to an event evidenced by Dongqi et al. (2022) [29], who observed an increase in micelle formation on microemulsion systems as the water/oil (W/O) ratio increased. Therefore, this phenomenon occurring during titration D may promote D-limonene protection.

When evaluating the composition of microemulsions, it becomes evident that the concentration of Labrasol in these formulations varies very little and does not exert a significant influence on the induction period, unlike the quantities of D-limonene and water. This phenomenon may be attributed to the application of a temperature of 120 °C during the assay. As demonstrated further in the thermal stress study, phase inversion of the microemulsion occurs from 45 °C onwards, potentially leading to the destabilization of formed micelles. In other words, the elevation of temperature within the assay chamber results in a higher solubility of Labrasol in the oil phase compared to the aqueous phase [30]. Within this context, a higher concentration of D-limonene in the formulation leads to a heightened susceptibility to micelle destabilization with temperature escalation, making D-limonene more accessible to oxidation, and consequently, reducing the induction period.

Furthermore, it is noteworthy to observe that the RapidOxy 100 equipment provided information about the microemulsions stability, as well their ability to protect D-limonene against volatilization, within a few hours of assay, in stark contrast to conventional studies that would require months for completion. Hence, it can be deduced that the assessment of oxidative stability emerges as an outstanding option for the characterization of microemulsions.

Although the initial investment in this methodology may exceed that of conventional stability chambers, it is essential to recognize that with the conventional technique, there is a monthly requirement for sample monitoring to draw conclusions regarding the product’s stability. This entails a larger sample quantity for analysis, as well as the availability of robust equipment (such as GC-MS) on a monthly basis to evaluate the oxidation trend of the product. In contrast, the oxidative stability technique employed allows for reliable results with a mere 5 g of sample, obtained in a matter of minutes. Considering these factors, it is conceivable that, in the end, the overall costs may align. Nevertheless, the cost-effectiveness of the oxidative stability approach is notably higher due to its efficiency and reduced resource demands.

##### Thermal Stress

The physical stability of D-limonene microemulsions were macroscopically evaluated after each heating cycle. It was observed that at 30 °C, all formulations remained stable without any alteration in their physical characteristics. However, formulation C4 began to show turbidity at 35 °C but returned to a normal state when removed from heating. At 40 °C, the C4 formulation became turbid again and showed slight phase separation, indicating that the system was disassembling. However, it became transparent and homogeneous when removed from heating. Formulation D5 also showed turbidity at this temperature.

In the next cycle, at 45 °C, formulations B4, C4, C5, and D5 became turbid. It should be noted that formulations B4, C4, and C5 have the highest concentrations of D-limonene in their composition (30.8%, 23.1%, and 21.4%, respectively), which may indicate that a higher content of this component is directly related to the lower physical stability of the formulation. This is because D-limonene is extremely volatile and, in higher concentrations, becomes more available for evaporation. Therefore, considering that microemulsion is a thermodynamically stable system resulting from the correct proportions of each component, the evaporation of D-limonene leads to an imbalance in the system, resulting in the rupture of microemulsion, which can be observed by turbidity and/or phase separation.

On the other hand, in addition to the volatilization of D-limonene, it is conceivable that the application of heat to the system might have induced a phase inversion phenomenon, which can be observed through the appearance of turbidity and/or phase separation in the emulsions due to alterations in surfactant affinities. For example, non-ionic surfactants such as Labrasol^®^ become more lipophilic under heating, in accordance with the observed outcomes for all formulations [31].

In addition, formulation D5 is the one with the highest concentration of Labrasol^®^ in its composition, so this component may be related to the turbidity observed at 45 °C, since storage recommendations for this surfactant include minimizing and controlling the degree of exposure to heat, light, and relative humidity [14,21].

From 50 °C, all formulations showed some level of turbidity, which remained until the temperature of 65 °C. At 70 °C, all formulations appeared to exhibit a phase transition followed by a possible separation of phases. However, after the decrease in temperature, all returned to the initial state of transparency and homogeneity, indicating that there was not actually a phase separation (irreversible), but the phenomenon of sedimentation (reversible) [32]. Thus, it can be concluded that from 50 °C, none of the formulations showed physical stability, which is probably related to the evaporation of D-limonene and/or phase inversion. However, under normal storage conditions (25 °C), it is expected that the formulations will remain stable, as they did not exhibit any actual instability at elevated temperatures.

In order to assess in greater detail the microemulsion’s ability to protect D-limonene against volatilization, analyses of oxidative stability (induction period), pH, and density were conducted following heating cycles, the results of which are described in Table 10.

According to the bidimensional plot (Figure 14), it is possible to visualize the separation of the samples into two principal groups, the first in the left side encompassing microemulsions D7 to D11 and a second on the right, with the formulations B4 and C4 to C7. The sum of both factors (PC1 and PC2) results in 96.22%, as an indication of the retention of a great part of the original variable information.

It is possible to observe that, in general, the induction period variation was more relevant than the other parameters, revealing that all formulations decreased after the thermal stress test. This is because part of the D-limonene probably oxidized or underwent some other type of reaction during the heating cycles, making the microemulsion less stable. However, it is noted that formulations with lower oil content remained more stable, with less pronounced differences in the induction period before and after thermal stress.

Therefore, it can be concluded that, despite the extreme conditions of agitation and heating to which they were exposed, the microemulsions proved to be suitable encapsulation techniques for the protection of D-limonene, increasing its resistance to higher temperatures.

### 3.2. D-limonene’s HLB Calculation

The LUMiSizer dispersion analyzer is used to evaluate the physical stability of suspensions and emulsions in a brief amount of time. This is possible by combining sample centrifugation with transmittance analysis within the cuvette length, making the analysis of particle sedimentation/creaming and emulsion stability possible [33].

The HLB was determined in a three-step process, in order to reduce the number of assays performed. Therefore, the HLB ranges were narrowed down in each step to determine an accurate value. The instability results of all steps are shown in Figure 15. In the first step, it was possible to observe that an HLB value of 13 resulted in the lowest instability index, consequently yielding higher formulation stability. In contrast, an HLB value of 15 led to the highest instability index, indicating a greater deviation of the D-limonene’s HLB from this value.

Subsequently, based on the first observations, HLB values between 12 and 14 were explored, with 12.6 notably possessing the lowest instability index and, thus, closest approximation to the HLB value of D-limonene. Finally, to precisely determine the HLB value of D-limonene, the range of HLB from 12.4 to 12.8 was examined, revealing that the HLB of D-limonene is equal to 12.5, attributed to its lower instability index (0.214).

## 4. Conclusions

Based on the results obtained, it can be concluded that the formation of microemulsions was possible using Labrasol^®^ as the sole surfactant (HLB = 12.0), which is consistent with the required HLB value of 12.5 determined for D-limonene. Therefore, it was determined that the microemulsion region in the ternary phase diagram includes concentrations of 7.1 to 30.8% D-limonene, 28.6 to 57.1% Labrasol^®^, and 20.0 to 64.3% water.

The formulations in question were homogeneous, transparent, low viscosity, and without visible particles, consistent with the characteristics of a microemulsion. Additionally, all formulations showed adequate D-limonene content, indicating that the room temperature production process was appropriate. The formulations exhibited good physical stability under mechanical stress (centrifugation), but all destabilized when subjected to a thermal stress test at 50 °C or higher.

Furthermore, the cost-effectiveness of the oxidative stability approach is notably higher due to its efficiency and reduced resource demands, providing reliable results regarding the behavior of D-limonene. It was possible to conclude that the oxidative stability of D-limonene was increased upon incorporation into the microemulsions. Thus, it can be inferred that the procedures employed were effective in obtaining microemulsions with desirable characteristics, and that this pharmaceutical form is a suitable encapsulation technique for the protection of D-limonene, increasing its resistance to higher temperatures.

## Figures and Tables

**Figure 1 pharmaceutics-15-02564-f001:**
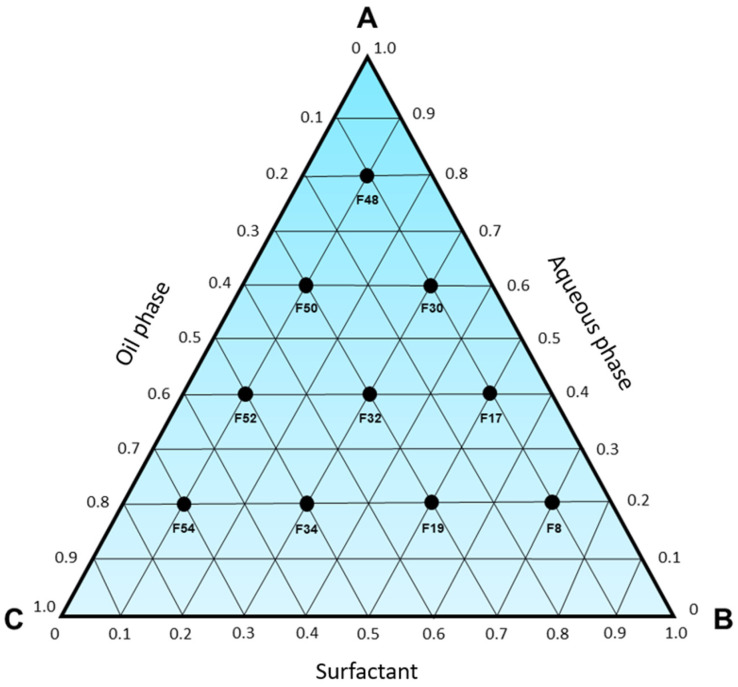
Pseudo-ternary phase diagram with 10 selected formulations for the characterization of the systems formed in each area of the triangle, where A—aqueous phase; B—surfactant; C—oil phase. The compositions of each point are described in Table 2.

**Figure 2 pharmaceutics-15-02564-f002:**
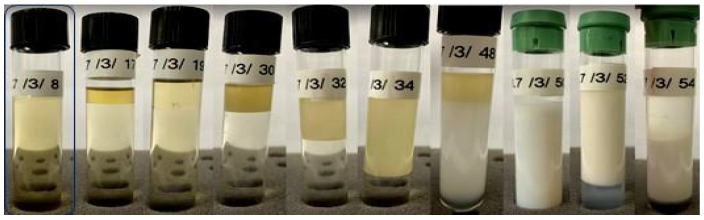
Formulations obtained based on an HLB required of 8.7, using a surfactant mixture of 67.65% Labrasol^®^ and 32.35% Span 85. The highlighted formulation is a microemulsion.

**Figure 3 pharmaceutics-15-02564-f003:**
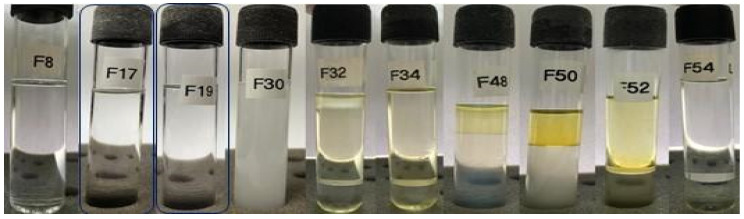
Formulations obtained based on an HLB requirement of 12, using 100% Labrasol^®^ as the surfactant. The highlighted formulations are microemulsions.

**Figure 4 pharmaceutics-15-02564-f004:**
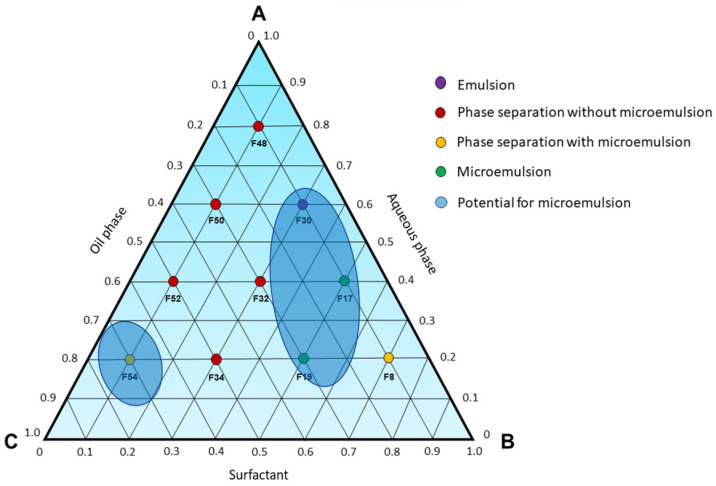
Ternary phase diagram (100% of Labrasol^®^ as surfactant), where A—aqueous phase; B—surfactant; C—oil phase. The compositions of each point are described in Table 2.

**Figure 5 pharmaceutics-15-02564-f005:**
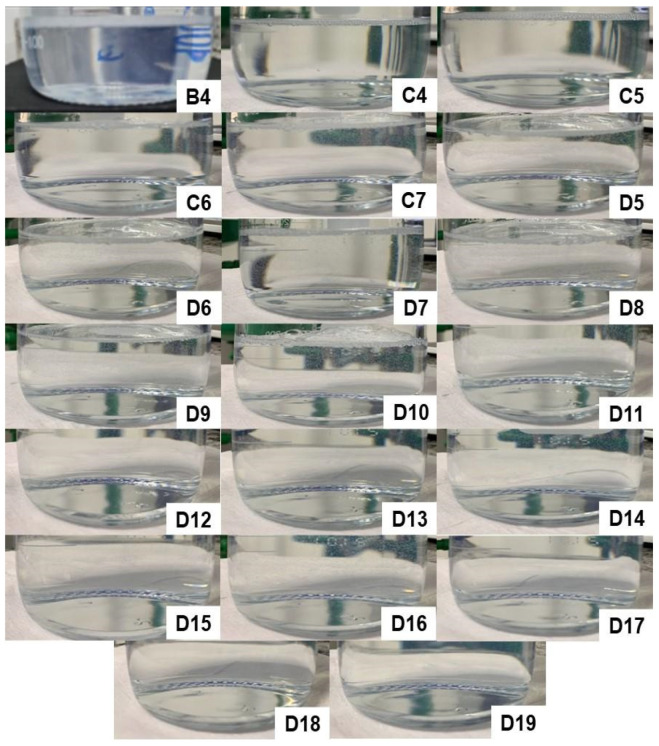
Microemulsions obtained through the titration technique, starting from the initial mixtures B, C, and D of D-limonene and surfactant.

**Figure 6 pharmaceutics-15-02564-f006:**
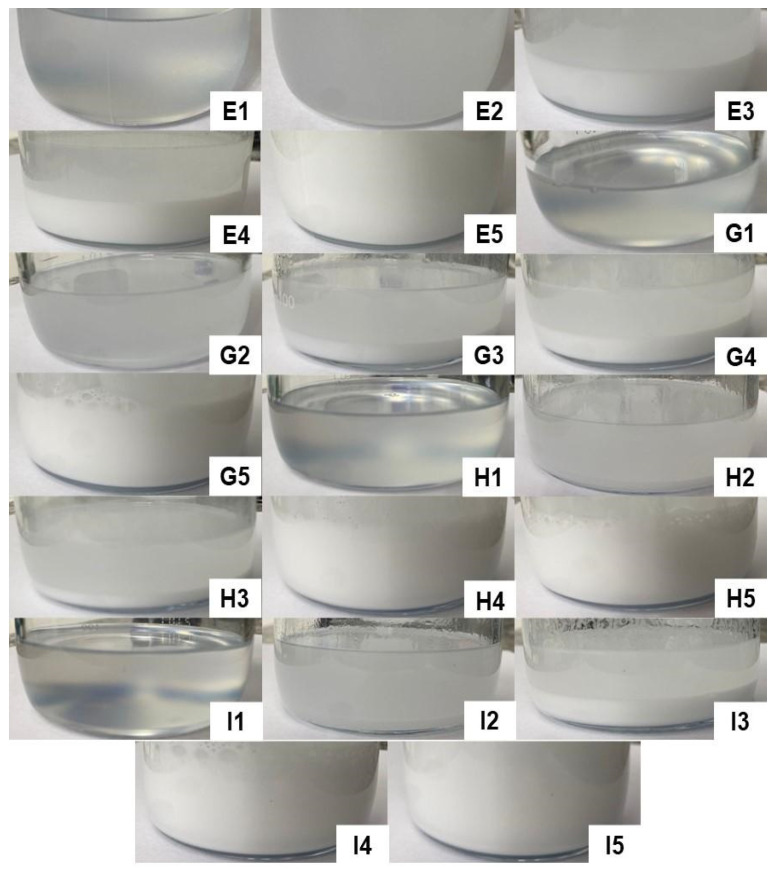
Formulations resulting from titrations E, G, H, and I, with none characterized as a microemulsion.

**Figure 7 pharmaceutics-15-02564-f007:**
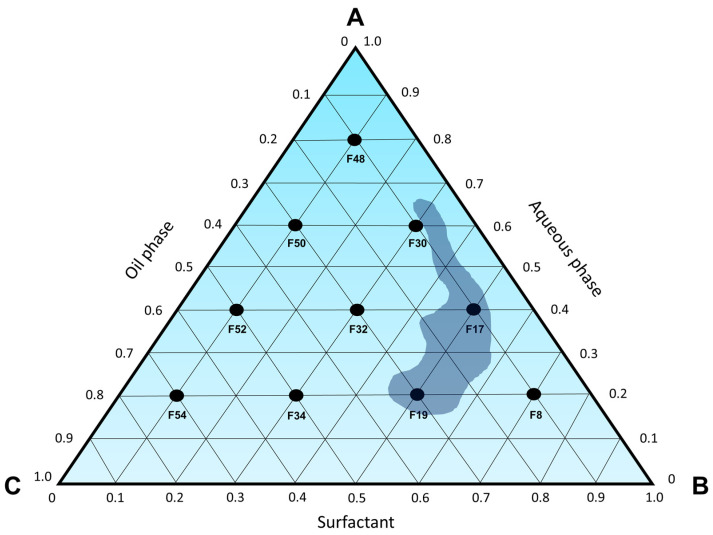
Microemulsions region (blue shape) obtained in the ternary phase diagram, in which A—water; B—Labrasol^®^; C—D-limonene.

**Figure 8 pharmaceutics-15-02564-f008:**
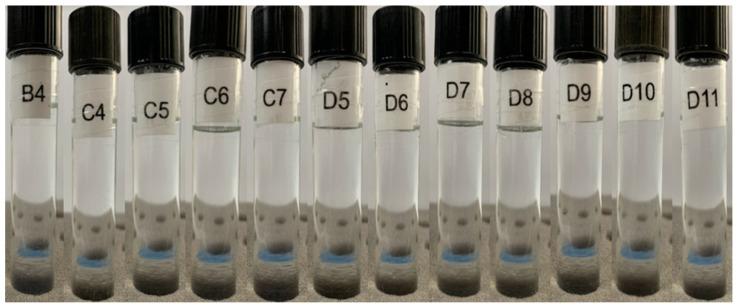
D-limonene microemulsions after 24 h of preparation. The compositions of each microemulsion (B4, C4, C5, C6, C7, D5, D6, D7, D8, D9, D10, and D11) are described in Table 7.

**Figure 9 pharmaceutics-15-02564-f009:**
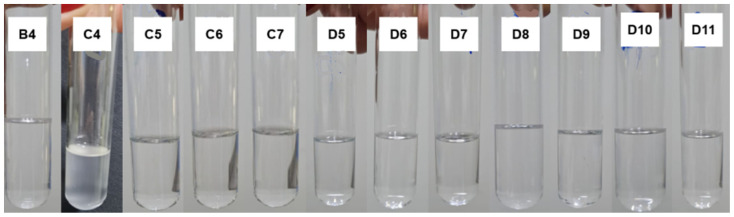
Formulations of D-limonene microemulsions after a 60 min cycle in a centrifuge at 4000 rpm. The compositions of each microemulsion (B4, C4, C5, C6, C7, D5, D6, D7, D8, D9, D10, and D11) are described in Table 7.

**Figure 10 pharmaceutics-15-02564-f010:**
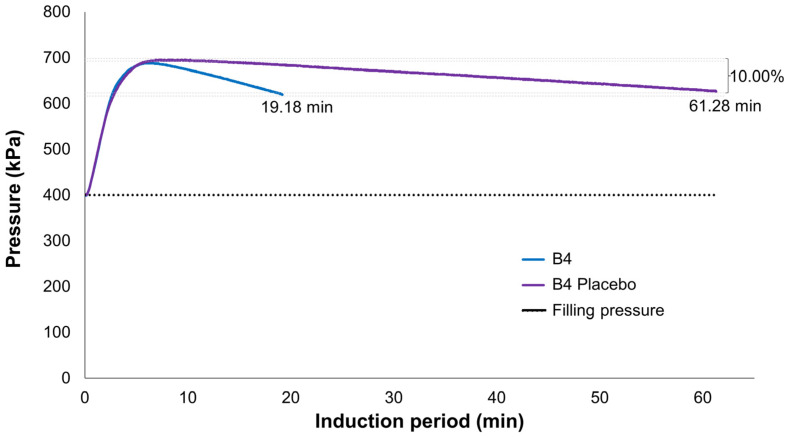
Comparison between the oxygen consumption profiles of a D-limonene microemulsion formulation and its corresponding placebo.

**Figure 11 pharmaceutics-15-02564-f011:**
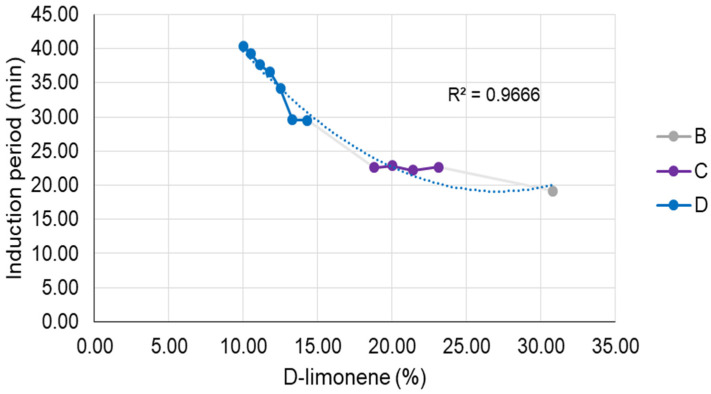
Correlation between D-limonene content and induction period for all titrations.

**Figure 12 pharmaceutics-15-02564-f012:**
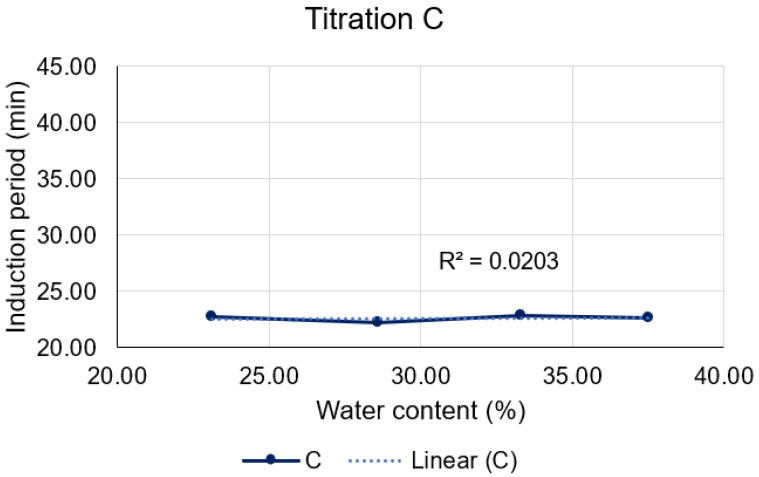
Correlation between water content and induction period for titration C (blue line), and its corresponding linear regression (dotted line).

**Figure 13 pharmaceutics-15-02564-f013:**
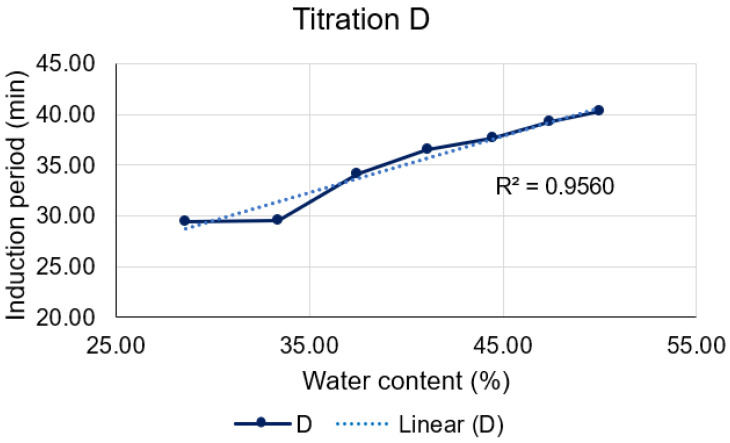
Correlation between water content and induction period for titration D (blue line), and its corresponding linear regression (dotted line).

**Figure 14 pharmaceutics-15-02564-f014:**
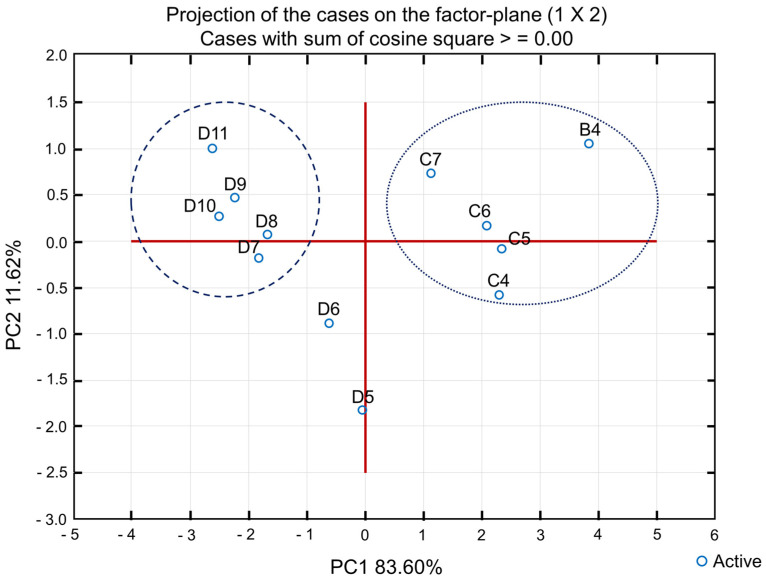
Bidimensional plot of principal component analysis from microemulsions characterization data.

**Figure 15 pharmaceutics-15-02564-f015:**
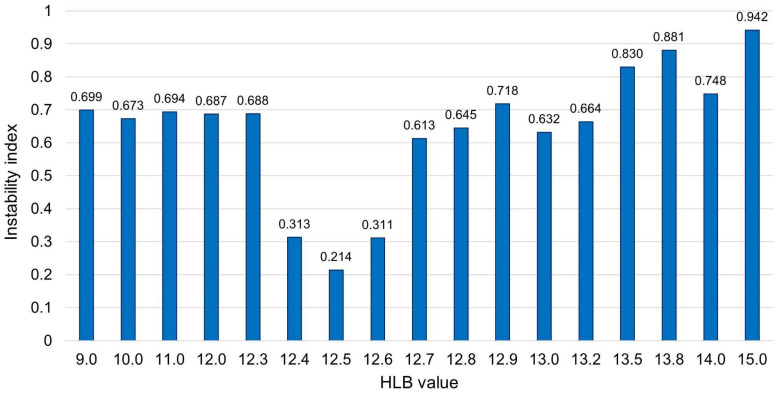
Instability index of D-limonene emulsion in a required HLB range of 9 to 15.

**Table 1 pharmaceutics-15-02564-t001:** Materials used in the development of D-limonene microemulsions.

Name	Supplier	Localization	Batch
D-limonene	Flavor Tec	Pindorama, Brazil	139/21
Labrasol^®^	Gattefossé	Barueri, Brazil	157233
Span 85-NV-LQ	Croda	Campinas, Brazil	00018779343
Span 80 Pharma-LQ	Croda	Campinas, Brazil	0001770301
Span 40 MBAL-PW	Croda	Campinas, Brazil	0001767384
Tween 60 NF MBAL-LQ	Croda	Campinas, Brazil	0001832549
Ultrapurified water	-	-	-

**Table 2 pharmaceutics-15-02564-t002:** Composition of the formulation selected in the pseudo-ternary phase diagram.

Formulation	Content (%)		
	Aqueous Phase	Oil Phase	Surfactant
8	20	10	70
17	40	10	50
19	20	30	50
30	60	10	30
32	40	30	30
34	20	50	30
48	80	10	10
50	60	30	10
52	40	50	10
54	20	70	10

**Table 3 pharmaceutics-15-02564-t003:** Surfactants and their respective HLB values according to the literature [13,14].

Commercial Name	Chemical Class	HBL
Labrasol^®^	Polyoxylglycerides (PEG-8 Caprylic/Capric Glycerides)	12.0
Span 85-NV-LQ	Sorbitan Trioleate	1.8
Span 80 Pharma-LQ	Sorbitan Monooleate	4.3
Span 40 MBAL-PW	Sorbitan monopalmitate	6.7
Tween 60 NF MBAL-LQ	Ethoxylated Sorbitan Ether	14.9

**Table 4 pharmaceutics-15-02564-t004:** Binary mixtures of surfactants to obtain the required HLB value.

Required HLB	Surfactant A		Surfactant B	
	Name	Concentration (%)	Name	Concentration (%)
6.4	Labrasol^®^	27.27	Span 80	72.73
	Labrasol^®^	45.10	Span 85	54.90
	Tween 60	19.81	Span 80	80.19
	Tween 60	35.11	Span 85	64.89
8.7	Labrasol^®^	37.74	Span 40	62.26
	Labrasol^®^	57.14	Span 80	42.86
	Labrasol^®^	67.65	Span 85	32.35
	Tween 60	24.39	Span 40	75.61
	Tween 60	41.51	Span 80	58.49
	Tween 60	52.67	Span 85	47.33
12	Labrasol^®^	100.00	-	-
	Tween 60	72.64	Span 80	27.36
	Tween 60	77.86	Span 85	22.14

**Table 5 pharmaceutics-15-02564-t005:** Concentration of the starting solutions for titration.

Formulation	D-limonene (%)	Labrasol^®^ (%)	Water Added per Step (mL)
A	55.0	45.0	10.0
B	40.0	60.0	10.0
C	30.0	70.0	10.0
D	20.0	80.0	10.0
E	75.0	25.0	10.0
G	80.0	20.0	10.0
H	90.0	10.0	10.0
I	95.0	5.0	10.0

**Table 6 pharmaceutics-15-02564-t006:** Surfactant proportions and respective HLB obtained.

HLB	Span^®^ 85	Labrasol^®^	Tween^®^ 80
9.0	0.45	-	0.55
10.0	0.38	-	0.62
11.0	0.30	-	0.70
12.0	0.23	-	0.77
12.3	-	0.90	0.10
12.4	-	0.87	0.13
12.5	-	0.83	0.17
12.6	-	0.80	0.20
12.7	-	0.77	0.23
12.8	-	0.73	0.27
12.9		0.70	0.30
13.0	0.15	-	0.85
13.2	-	0.60	0.40
13.5	-	0.50	0.50
13.8	-	0.40	0.60
14.0	0.08	-	0.92
15.0	-	-	1.00

**Table 7 pharmaceutics-15-02564-t007:** D-limonene microemulsion composition.

Formulation	D-limonene (%)	Labrasol^®^ (%)	Water (%)
B4	30.8	46.2	23.1
C4	23.1	53.8	23.1
C5	21.4	50.0	28.6
C6	20.0	46.7	33.3
C7	18.8	43.8	37.5
D5	14.3	57.1	28.6
D6	13.3	53.3	33.3
D7	12.5	50.0	37.5
D8	11.8	47.1	41.2
D9	11.1	44.4	44.4
D10	10.5	42.1	47.4
D11	10.0	40.0	50.0
D12	9.5	38.1	52.4
D13	9.1	36.4	54.5
D14	8.7	34.8	56.5
D15	8.3	33.3	58.3
D16	8.0	32.0	60.0
D17	7.7	30.8	61.5
D18	7.4	29.6	63.0
D19	7.1	28.6	64.3

**Table 8 pharmaceutics-15-02564-t008:** Average content in (± standard deviation) and mean pH values (± standard deviation, *n* = 3) of D-limonene microemulsions.

Formulation	Average Content (%)	Mean pH	Mean Density (g/cm^3^)
B4	100.649 ± 0.949	3.55 ± 0.07	0.9844 ± 0.0005
C4	100.622 ± 0.120	3.58 ± 0.03	1.0016 ± 0.0001
C5	99.554 ± 0.130	3.55 ± 0.05	1.0006 ± 0.0013
C6	105.000 ± 0.000	3.53 ± 0.06	1.0002 ± 0.0003
C7	100.621 ± 0.295	3.47 ± 0.05	1.0006 ± 0.0002
D5	100.009 ± 0.846	3.60 ± 0.06	1.0199 ± 0.0005
D6	102.709 ± 0.552	3.47 ± 0.04	1.0186 ± 0.0007
D7	101.974 ± 0.968	3.42 ± 0.01	1.0186 ± 0.0002
D8	100.826 ± 0.235	3.41 ± 0.03	1.0168 ± 0.0007
D9	101.553 ± 0.662	3.39 ± 0.02	1.0159 ± 0.0006
D10	104.029 ± 0.343	3.43 ± 0.04	1.0160 ± 0.0002
D11	104.303 ± 0.240	3.39 ± 0.03	1.0129 ± 0.0003

**Table 9 pharmaceutics-15-02564-t009:** Results of the oxidative stability of D-limonene based on induction period.

Sample	Induction Period (min)
D-limonene	4.88
B4	19.18
B4 placebo	61.28
C4	22.67
C4 placebo	36.83
C5	22.18
C5 placebo	41.17
C6	22.83
C6 placebo	47.57
C7	22.57
C7 placebo	49.93
D5	29.48
D5 placebo	36.02
D6	29.55
D6 placebo	46.10
D7	34.17
D7 placebo	57.37
D8	36.58
D8 placebo	74.53
D9	37.65
D9 placebo	58.17
D10	39.30
D10 placebo	80.98
D11	40.33
D11 placebo	67.88

**Table 10 pharmaceutics-15-02564-t010:** Results of induction period, pH ± standard deviation (*n* = 3), and density ± standard deviation (*n* = 3) of microemulsions after thermal stress.

Formulation	Induction Period (min)	Mean pH	Mean Density (g/cm^3^)
B4	16.97	3.45 ± 0.04	0.9829 ± 0.0003
C4	20.13	3.40 ± 0.02	0.9989 ± 0.0008
C5	19.98	3.39 ± 0.03	0.9943 ± 0.0006
C6	20.53	3.36 ± 0.02	0.9944 ± 0.0008
C7	23.32	3.18 ± 0.01	0.9973 ± 0.0012
D5	25.97	3.24 ± 0.02	1.0140 ± 0.0018
D6	23.05	3.17 ± 0.01	1.0138 ± 0.0006
D7	30.90	3.13 ± 0.01	1.0157 ± 0.0003
D8	27.70	3.10 ± 0.01	1.0125 ± 0.0005
D9	32.78	3.03 ± 0.02	1.0124 ± 0.0006
D10	37.53	3.02 ± 0.02	1.0137 ± 0.0004
D11	38.77	3.01 ± 0.01	1.0104 ± 0.0004

## Data Availability

The data presented in this study are available in the article and in the Appendix A.

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
