# Peer review of "Microemulsions: An Encapsulation Strategy to Increase the Thermal Stability of D-limonene"

_pharmaceutics, 2023, doi:10.3390/pharmaceutics15112564_

Round 1

Reviewer 1 Report

The research work “Microemulsions: an encapsulation strategy to increase the thermal stability of volatile substances” is novel and necessary looking towards industrial demand. The observations and comments are as follows:

1.       Introduction: Literature of other methods/work done till data and their limitation if any. Addition of statement, How this work is different than that.

2.       Method: Quantitative analysis, solvent system not given. (Even in the Supplement file it's missing)

3.       2.2.4.7. Thermal stress, Why temperature range is 30 to 70? As max temperature worldwide is around 50.  Is there any specific need to give higher temperature stress?

4.       I would like to see the zeta potential data and its effect on stability. Please include zeta potential data.

5.       Line 213..and onwards: 3.1. Definition of surfactant. No need for this section and discussion on HLB in so much detail. One or two sentences may be enough. The heading is not appropriate either.

6.       Figure 2-14 are identical. Expecting only figure of optimized or final combination, rest all move to supplement files. Follow Similarly for Figure 16-23.

7.       Why do authors measure the pH and density?

8.  Combine the titration data and make it one figure for better comparison.

9.       Merge figures 30-32. Why SD is missing?

10.   Conclusion: Can the author talk about the rise in cost compared to simple stabilizing methods?

Overall: The work is important as far as industrial application is there. However, the representation and finding of data are not sufficient to provide significant novelty and proof of concept.

Author Response

Firstly, we would like to extend our heartfelt thanks to the reviewer, whose significant input greatly enhanced the adaptation of our article to meet the standards set by Pharmaceutics. Recognizing the commendable efforts of this reviewer, we promptly incorporated their suggestions. Presented below are our responses and supporting arguments.

  1. Introduction: Literature of other methods/work done till data and their limitations if any. Addition of statement, how is this work different from that?

To address this suggestion, the introduction of the manuscript has been rewritten, and the following new references have been added:

  1. KARADAÄž, AyÅŸe Esra; OKUR, Neslihan ÜstündaÄŸ; DEMIRCI, Betül; DEMIRCI, Fatih. Rosmarinus officinalis L. Essential Oil Encapsulated in New Microemulsion Formulations for Enhanced Antimicrobial Activity. J. Surfactants Deterg. 2021, 25(1), 95-103. Wiley. http://dx.doi.org/10.1002/jsde.12549.
  2. TOLEDO, et al. Improved in vitro and in vivo Anti-Candida albicans Activity of Cymbopogon nardus Essential Oil by Its Incorporation into a Microemulsion System. Int. J. Nanomedicine 2020, 15, 10481-10497. Informa UK Limited. http://dx.doi.org/10.2147/ijn.s275258.
  3. LAOTHAWEERUNGSAWAT, Natnaree et al. Transdermal Delivery Enhancement of Carvacrol from Origanum vulgare L. Essential Oil by Microemulsion. Int. J. Pharm. 2020, 579, 119052. Elsevier BV. http://dx.doi.org/10.1016/j.ijpharm.2020.119052.
  4. MEHANNA, Mohammed M.; ABLA, Kawthar Khalil; ELMARADNY, Hoda A. Tailored Limonene-Based Nanosized Microemulsion: Formulation, Physicochemical Characterization and In-Vivo Skin Irritation Assessment. Adv. Pharm. Bull. 2020, 11(2), 274-285. Maad Rayan Publishing Company. http://dx.doi.org/10.34172/apb.2021.040.

These changes can be found in lines 49 to 59 and, subsequently, in lines 69 to 75. Also, we would like to point out that the primary distinction of our work lies in the utilization of oxidative stability as a tool for characterizing microemulsions, particularly those containing volatile substances. We have not come across any other study employing this technique. Despite its limitations, such as not elucidating the degradation products of D-limonene, it allows for an indirect assessment of the stability of D-limonene when incorporated into a microemulsion. Therefore, in cases where researchers may not have access to more expensive techniques like GC-MS, the use of oxidative stability techniques provides reliable and robust insights into the stability of the developed product, especially with regard to the evaporation of D-limonene. 

Another crucial aspect is the utilization of a dispersion analyzer to ascertain the EHL value of D-limonene, a contentious piece of information in the existing literature. With the aid of this technique, we were able to accurately determine the EHL of D-limonene within a matter of minutes. This approach has significantly contributed to a more precise understanding of the compound's behavior.

  1. Method: Quantitative analysis, solvent system not given. (Even in the Supplement file it's missing)

We apologize for not initially providing this information. The necessary changes have been made throughout the text of the article (line 166 and supplementary material), clarifying that the solvent system used was absolute ethanol.

  1. 2.2.4.7. Thermal stress, Why temperature range is 30 to 70? As max temperature worldwide is around 50.  Is there any specific need to give higher temperature stress?

The present paper is part of a PhD thesis, whose objective is to incorporate the microemulsion into liquisolid pellets. Therefore, it is imperative that the microemulsions are capable of withstanding the inherent drying process involved in pellet production. In order to further elucidate this point for the readers, we have included information that a thermal stress study was conducted up to 70°C to evaluate the formulation's ability to resist industrial processes (lines 212 and 213).

  1. I would like to see the zeta potential data and its effect on stability. Please include zeta potential data.

We understand and agree about the significance of assessing the zeta potential in microemulsion characterization. However, regrettably, we do not have the technology available for such a study. Nonetheless, we believe that the characterization tests we have conducted allow us to conclude that the obtained microemulsions exhibit good mechanical, thermal, and oxidative stability, making them suitable for our intended purposes.

  1. Line 213 and onwards: 3.1. Definition of surfactant. No need for this section and discussion on HLB in so much detail. One or two sentences may be enough. The heading is not appropriate either.

The topic has been re-evaluated, and we fully concur with the reviewer. Consequently, the topic has been removed from the paper's text.

  1. Figure 2-14 are identical. Expecting only a figure of optimized or final combination, rest all move to supplement files. Follow Similarly for Figure 16-23.

Firstly, we would like to clarify that although the figures may appear similar, they represent all the formulations obtained from various surfactant mixtures, consequently yielding different HLB values, and providing the reader with an insight into the researchers' workbench. Besides that, we agree that an excess of images is unnecessary. As a result, we have retained only those where one or more microemulsions were obtained (Figure 2 and 3), while relocating the remaining images to the supplementary material.

  1. Why do authors measure the pH and density?

The pH and density measurements were conducted with the aim of providing additional insights into the comparative parameters of similarity between the formulations. This enabled us to perform a multivariate statistical analysis, enhancing the data analysis for the study. For enhanced reader comprehension, we have added the following sentence at lines 436 and 437: "In order to obtain additional comparative parameters of similarity between the formulations, the microemulsions were also characterized in terms of pH and density."

  1. Combine the titration data and make it one figure for better comparison.

In order to comply with this request, we have exclusively retained images depicting the successful acquisition of microemulsions (Figure 5). Additionally, we have consolidated all images from titrations E, G, H, and I into a single figure (Figure 6). Subsequently, we have relocated the remaining images to the supplementary material.

  1. Merge figures 30-32. Why is SD missing?

We agree that consolidating all the figures resulting from the study on the determination of the HLB of D-limonene leads to a better understanding. This request has been promptly addressed (Figure 15). Regarding the standard deviation value, the LUMiSizer equipment measures the transmittance profile of the samples. Based on the 800 measurements taken, the SEPView® 6 software calculates the instability index, but, unfortunately, it does not provide a standard deviation value for this specific parameter.

  1. Conclusion: Can the author talk about the rise in cost compared to simple stabilizing methods?

This is an important discussion, and the following paragraph has been added to the topic "3.1.3.4 Oxidative stability" (lines 584 - 593) to elucidate that: “Although the initial investment in this methodology may exceed that of conventional stability chambers, it is essential to recognize that with the conventional technique, there is a monthly requirement for sample monitoring to draw conclusions regarding the product's stability. This entails a larger sample quantity for analysis, as well as the availability of robust equipment (such as GC-MS) on a monthly basis to evaluate the oxidation trend of the product. In contrast, the oxidative stability technique employed allows for reliable results with a mere 5 g of sample, obtained in a matter of minutes. Considering these factors, it is conceivable that, in the end, the overall costs may align. Nevertheless, the cost-effectiveness of the oxidative stability approach is notably higher due to its efficiency and reduced resource demands.

Furthermore, the following sentence has also been incorporated into the topic 5. Conclusions (lines 696 - 698): “the cost-effectiveness of the oxidative stability approach is notably higher due to its efficiency and reduced resource demands, providing reliable results regarding the behavior of D-limonene.

Overall: The work is important as far as industrial application is there. However, the representation and finding of data are not sufficient to provide significant novelty and proof of concept.

We sincerely appreciate your thoughtful consideration of the industrial significance of our work. We have carefully taken into account your feedback regarding the representation and findings of our data, incorporating all the suggestions provided during the initial review. After conducting an extensive analysis, we firmly believe that the presented data sufficiently support our conclusions and demonstrate the feasibility of our approach.

Also, the manuscript presents a comprehensive report on the development and physicochemical characterization of D-limonene microemulsion formulations, offering valuable insights into the process of planning and obtaining microemulsions of volatile substances, the selection of appropriate surfactants, and the potential of this pharmaceutical form to function as a suitable delivery system for D-limonene. 

Moreover, the manuscript introduces an innovative technology, namely oxidative stability, for evaluating the protection against evaporation provided by microemulsions, which obviates the need for costly analytical methodologies such as GC-MS (gas chromatography-mass spectrometry). Considering these contributions, we anticipate that our manuscript will garner significant interest and citations in the relevant literature.

Once again, we extend our gratitude for your valuable insights and feedback.

Kind regards.

Reviewer 2 Report

 ·         The manuscript titled "Microemulsions: An Encapsulation Strategy for Enhancing the Thermal Stability of Volatile Substances" is a well-written article, and I would like to highlight some key observations.

·         In the title, the authors use the term "stability of volatile substances," but they exclusively focus on D-limonene as the volatile substance. It may not be appropriate to generalize their findings to all volatile substances. Therefore, I recommend replacing the term "stability of volatile substances" with "the stability of D-Limonene."

·         Regarding the Oxidative Stability Assessment: The main objective of this test is to assess how well the formulation resists oxidation under accelerated conditions. It remains unclear whether D-limonene has consumed the oxygen or if other components of the formula have undergone oxidation. Since the conditions of this experiment are quite severe, it would be beneficial to analyze the formula by distilling the D-limonene and subjecting it to GC-MS analysis. Among the potential byproducts of D-limonene, limonene oxide, carvone, perillyl alcohol, and carveol could be considered.

·         While the authors conducted a centrifugation test, I would like to inquire about the stability of the formula under vibration and agitation conditions. This is important for simulating real-world scenarios that cosmetic products may encounter during transportation, handling, and everyday use.

·         Furthermore, the formula appears to be stable under standard room temperature conditions and mechanical stress, but it shows instability at higher temperatures (50°C or above). So what is the expected shelf-life of your formula under both normal and drastic storage conditions?

Author Response

Firstly, we would like to extend our heartfelt thanks to the reviewer, whose significant input greatly enhanced the adaptation of our article to meet the standards set by Pharmaceutics. Recognizing the commendable efforts of this reviewer, we promptly incorporated their suggestions. Presented below are our responses and supporting arguments.

  1. In the title, the authors use the term "stability of volatile substances," but they exclusively focus on D-limonene as the volatile substance. It may not be appropriate to generalize their findings to all volatile substances. Therefore, I recommend replacing the term "stability of volatile substances" with "the stability of D-Limonene."

We have reviewed the work and fully agree with the suggestion made. Therefore, we have amended the title of the article to "Microemulsions: an encapsulation strategy to increase the thermal stability of D-limonene."

  1. Regarding the Oxidative Stability Assessment: The main objective of this test is to assess how well the formulation resists oxidation under accelerated conditions. It remains unclear whether D-limonene has consumed the oxygen or if other components of the formula have undergone oxidation. Since the conditions of this experiment are quite severe, it would be beneficial to analyze the formula by distilling the D-limonene and subjecting it to GC-MS analysis. Among the potential byproducts of D-limonene, limonene oxide, carvone, perillyl alcohol, and carveol could be considered.

We acknowledge that the presentation of the results may have raised questions regarding the oxidation of the formulation, and it may not have been entirely clear whether it pertained to D-limonene or other components. To address this concern, we have included a new graph (Figure 10) in the manuscript, and the following paragraph in Lines 510 - 517: “Figure 10 provides an example of the observed difference between the formulation and its corresponding placebo, indicating that Labrasol® oxidation in water starts only after the oxygen consumption of the microemulsion is completed. Thus, it is understood that oxidation observed in microemulsions is mostly, if not exclusively, due to D-limonene. Although it may not be possible to identify D-limonene byproducts such as limonene oxide, carvone, and perillyl alcohol through this technique [30], it can serve as an alternative and indirect means to assess both the oxidation and volatilization of D-limonene in the absence of more robust techniques like GC-MS.

Furthermore, we fully appreciate the significance of conducting an analysis via GC-MS. Regrettably, we do not currently have access to this technology for the analysis of D-limonene degradation products. In this context, we find that the oxidative stability technique has proven to be an alternative and effective means of evaluating, albeit indirectly, the resistance of D-limonene to evaporation and oxidation, particularly within the context of its incorporation into a microemulsion.

  1. While the authors conducted a centrifugation test, I would like to inquire about the stability of the formula under vibration and agitation conditions. This is important for simulating real-world scenarios that cosmetic products may encounter during transportation, handling, and everyday use.

We believe that the results obtained during centrifugation can be extrapolated for an analysis of vibration and agitation. In order to clarify that, the following paragraph were added in section 3.1.3.3 Centrifugation (lines 467 - 472): “Based on these findings, it is possible to infer that the microemulsions exhibit good mechanical stability, such that the applied stressful conditions of agitation and vibrations were unable to disrupt the system. Even the formulation C4 that showed higher sensitivity to agitation returned to thermodynamic equilibrium a few minutes after the cessation of agitation. Therefore, it is highly likely that these products are capable of withstanding normal conditions of transportation and handling, for example.

  1. Furthermore, the formula appears to be stable under standard room temperature conditions and mechanical stress, but it shows instability at higher temperatures (50°C or above). So what is the expected shelf-life of your formula under both normal and drastic storage conditions?

Thank you for this observation, as it has allowed us to reevaluate the results and rephrase the paragraph as follows (lines 625 to 633): “At 70°C, all formulations appeared to exhibit a phase transition followed by a possible separation of phases. However, after the decrease in temperature, all returned to the initial state of transparency and homogeneity, indicating that there was not actually a phase separation (irreversible), but the phenomenon of sedimentation (reversible) [34]. Thus, it can be concluded that from 50°C, none of the formulations showed physical stability, which is probably related to the evaporation of D-limonene and/or phase inversion. However, under normal storage conditions (25°C), it is expected that the formulations will remain stable, as they did not exhibit any actual instability at elevated temperatures.

Once again, we extend our gratitude for your valuable insights and feedback.

Kind regards.

Reviewer 3 Report

Please find my comments on your manuscript " Microemulsions: an encapsulation strategy to increase the ther- 2mal stability of volatile substances"

·       The title mentions thermal stability, but this variable is not mentioned in the abstract. Please revise the abstract to include the main objectives and findings related to thermal stability.

·        In Table 1, please specify the city and country of the supplier for each material.

•  In Figure 1, please label the sides of the triangle with the corresponding variables (oil, water, and surfactant).

•  Please explain how Table 2 and Table 4 are connected in practice. Which composition of binary surfactant was used in the preparation of each formulation? Please provide complete details.

•  Please elaborate more on section 2.2.3.1. Why was a water-in-oil emulsion prepared? What was the purpose of preparing 100 ml of emulsion? Why was the section titled "titration method"?

•  In section 2.2.4.1., how did you use the UV method to determine the concentration of D-limonene in the emulsion, considering that other components such as surfactant could also absorb UV light?

•  Table 6 should indicate the molar fraction of each component if it was used in the calculations.

•  In Figure 8, as stated in line 269 of the text, please highlight the best formulation in the figure.

•  In figures 2 to 14, please indicate which systems are microemulsions.

•  As stated in lines 301 to 305 of the text, "Labrasol® is likely associated with the formation of microemulsion". However, since the optimal formulations have more than 50% surfactant, it seems that they form "micelle systems" instead. It seems that if you use a "micelle system" approach, you would need fewer samples in the development process. How do you justify this theory? Generally, when more oil phase than surfactant is needed, an emulsion system is formed, and when more surfactant is needed to form a spontaneous microemulsion system, a "micelle system" is formed.

•  It is recommended to combine figures 2 to 13 into two or three figures with appropriate grouping based on similar characteristics or trends.

•  It is recommended to combine figures 17 to 23 into two or three figures with appropriate grouping based on similar characteristics or trends.

•  It is strongly recommended to use the definition of micelle in describing section 3.2.2 Titration method and explain how it relates to the formation and stability of microemulsions.

•  In Figures 17 to 23, the formulation components are unclear. Please provide more explanation and refer to them in the figure captions.

•  Considering the molecular structure of Labrasol®, it seems that this molecule is prone to oxidation. How was accounted for in the oxidative stability test?

•  As the optimal formulations have nearly 50% surfactant, it is strongly recommended to discuss the correlation between micelle formation and stability in a separate section in the text.

·        The number of references used in this study is insufficient. It is recommended to review and cite relevant and valid studies in the discussion section to support the findings and compare them with the existing literature.

Regards, 

Author Response

Firstly, we would like to extend our heartfelt thanks to the reviewer, whose significant input greatly enhanced the adaptation of our article to meet the standards set by Pharmaceutics. Recognizing the commendable efforts of this reviewer, we promptly incorporated their suggestions. Presented below are our responses and supporting arguments.

  1. The title mentions thermal stability, but this variable is not mentioned in the abstract. Please revise the abstract to include the main objectives and findings related to thermal stability.

  We apologize for the initial lack of clarity in this information. In order to provide clarification, the abstract has been revised to incorporate the following sentences at lines 9 - 12 and 21 - 23, respectively: “D-limonene, derived from citrus essential oils, holds significant therapeutic potential but faces challenges due to its high volatility, especially in pharmaceutical formulations. This study investigates microemulsions as a promising delivery system for volatile compounds, emphasizing their thermal protection for D-limonene.”, and “Notably, the induction period of the microemulsions remained practically unchanged pre and post-heating (70°C), suggesting the formulation's ability to enhance the D-limonene thermal stability.”.

  1. In Table 1, please specify the city and country of the supplier for each material.

  To address this request, the column "localization" has been added to Table 1 (line 85).

  1. In Figure 1, please label the sides of the triangle with the corresponding variables (oil, water, and surfactant).

  To comply with this request, all figures containing a pseudoternary phase diagram (Figure 1, Figure 4, and Figure 7) have been supplemented with the labels "aqueous phase," "surfactant," and "oil phase" on each side of the triangle.

  1. Please explain how Table 2 and Table 4 are connected in practice. Which composition of binary surfactant was used in the preparation of each formulation? Please provide complete details.

  We would like to clarify that the 10 formulations described in Table 2 were produced 13 times, with the surfactant being altered in each instance. In each case, the surfactant used corresponds to the mixtures described in Table 4. In order to provide greater clarity on the methodology and approach for the reader, we have included the following paragraph in Section 2.2.2 Definition of surfactants (lines 111 - 119): “In order to determine the appropriate excipient for the formulation, a comprehensive evaluation was conducted involving a range of materials (Table 3), either individually or in binary combinations. The surfactant concentrations provided in Table 2 corresponded to the constituents of each binary combination and/or single surfactant as delineated in Table 4. In other words, each combination was utilized in the production of the ten formulations outlined in Table 2, resulting in a total of 130 distinct formulations. This expansive array was established to delineate the microemulsion region associated with each surfactant, thereby facilitating the identification of the most suitable excipient, along with its optimal concentration.

  1. Please elaborate more on section 2.2.3.1. Why was a water-in-oil emulsion prepared? What was the purpose of preparing 100 ml of emulsion? Why was the section titled "titration method"?

  In order to address all the points raised, section 2.2.3.1 Titration Method has been rewritten as follows: “Once the formulations with potential for microemulsion formation were identified, a region of the ternary phase diagram was delimited and explored by the titration method. This method involves the preparation of a mixture of surfactant and oil phase, with known proportions, followed by the incremental addition of small quantities of water, mimicking an analytical titration. Following each water addition, a simple agitation was performed, and subsequently, the resulting product was macroscopically characterized to ascertain the presence or absence of microemulsion formation [17]. To explore the microemulsion areas, 8 different surfactant and D-limonene proportions were used (Table 5), starting from 100 mL of each mixture for better system visualization within a Schott flask.

 Given the precise measurement of the added water, it is feasible to calculate the proportions of the three components of the formulation at each step, elucidating the pseudoternary phase diagram with respect to the microemulsion regions.

  1. In section 2.2.4.1., how did you use the UV method to determine the concentration of D-limonene in the emulsion, considering that other components such as surfactant could also absorb UV light?

  Firstly, we would like to clarify that this manuscript is part of an ongoing PhD thesis, with the development of the analytical method for quantifying D-limonene via UV being a key aspect, with unpublished data. However, to complement the current work, the following sentences have been included in the methodology 2.2.4.1 D-limonene content (lines 166 - 170): "Subsequently, the microemulsions were quantified using an Evolution 201 spectrophotometer (Thermo Fisher Scientific, USA) at 280 nm. This wavelength was selected due to the absence of signal interference from Labrasol® at its highest concentration in the formulation, as well as the absence of signal from absolute ethanol as the solvent system (Figures S1 and S2)."

Additionally, in the supplementary material, scans of the placebo and solvent system were included to demonstrate that there is no signal from any of the components at the selected wavelength.

  1. Table 6 should indicate the molar fraction of each component if it was used in the calculations.

  The methodology employed for calculating the quantity of each surfactant in the mixture to achieve the required HLB was based on Equations 1 and 2, without the utilization of molar fractions. This clarification has been provided in lines 237 - 239 with the following statement: "This system included either Span 85 or Labrasol® combined with Tween 80, in varying proportions, in order to achieve the required HLB value (Table 6), also calculated by Equations 1 and 2."

  1. In Figure 8, as stated in line 269 of the text, please highlight the best formulation in the figure.

  In order to address the suggestions made during the manuscript review, the figure referred to by the reviewer as Figure 8 has been renumbered to Figure 2. In this case, the microemulsion formulation obtained has been highlighted with a blue circle in the image.

  1. In figures 2 to 14, please indicate which systems are microemulsions.

  As mentioned in the previous comment, during the manuscript review, we have retained only those figures where one or more microemulsions were obtained (Figure 2 and 3), while relocating the remaining images to the supplementary material. In both cases, the microemulsions are highlighted with blue circles in the image.

  1. As stated in lines 301 to 305 of the text, "Labrasol® is likely associated with the formation of microemulsion". However, since the optimal formulations have more than 50% surfactant, it seems that they form "micelle systems" instead. It seems that if you use a "micelle system" approach, you would need fewer samples in the development process. How do you justify this theory? Generally, when more oil phase than surfactant is needed, an emulsion system is formed, and when more surfactant is needed to form a spontaneous microemulsion system, a "micelle system" is formed.

  First and foremost, we would like to acknowledge your feedback. However, we'd like to provide some further context for our choice of focusing on microemulsions, based on the following points:

  1. While it is widely known that microemulsions typically form spontaneously with high concentrations of surfactants, sometimes necessitating the addition of a co-surfactant (AZAD et al. 2020; PAVONI et al. 2020), we opted for the approach of exploring the pseudoternary phase diagram. This decision was made with the aim of gaining a deeper understanding of all the systems achievable with different concentrations of surfactant, D-limonene, and water, with a clear emphasis on the microemulsion region. Furthermore, we would like to emphasize that it was possible to achieve microemulsions within a concentration range of 28.6% to 57.1% of Labrasol. This means that the desired pharmaceutical form was attained even with lower concentrations of surfactant, confirming that we were able to explore all possible formulation options of microemulsions with water, D-limonene, and Labrasol to their fullest extent.
  2. The observation in the text that "Labrasol® is likely associated with the formation of microemulsion" is indeed significant. We would like to highlight that what we intended to convey with this statement was that Labrasol's molecular structure likely played a pivotal role in obtaining the microemulsions. This is evidenced by the fact that other surfactants, even at concentrations similar to Labrasol, did not yield the same level of success. Therefore, in this specific case, the structure of Labrasol seemed to be more influential than the surfactant concentration itself. To clarify this point further, we have made the following alteration in the manuscript (lines 285 - 295): “Labrasol®, categorized as caprylocaproyl polyoxylglycerides, represents a non-ionic surfactant with primary utility as a solubilizing agent, a surfactant for microemulsions, and a lightweight foam generator when used in conjunction with pump devices, obviating the necessity for propellants. Its molecular composition encompasses a minor proportion of mono-, di-, and triglycerides, with a predominant presence of PEG-8 (MW 400), along with mono- and diesters derived from caprylic (C8) and capric (C10) acids [14; 20]. The distinctive structural configuration of Labrasol®, divergent from the other evaluated surfactants, is presumably instrumental in the achievement of microemulsions. This is substantiated by the inability of Span 85-NV-LQ, Span 80 Pharma-LQ, Span 40 MBAL-PW, and Tween 60 NF MBAL-LQ to facilitate microemulsion formation, even at elevated concentrations.
  3. We understand that adopting a micelle system approach would likely require fewer experiments, as in micellar dispersions, the aggregates primarily comprise surfactant and are typically dispersed in water (LANGEVIN, D. 1992). However, it is important to note that while microemulsions can be conveniently likened to micelles (TADROS, 2013), the aggregates in microemulsions are substantially larger. They encompass sizable liquid cores (oil in water or water in oil) surrounded by a monolayer of surfactant that stabilizes the dispersion (LANGEVIN, D. 1992). Although micelles can be swollen to a limited extent by oils, achieving the high degree of swelling observed in microemulsions necessitates the use of suitable surfactants, as exemplified by Labrasol in our study.
  4. It is also widely acknowledged that the curvature and consequently, the sizes of micelles obtained in a micellar system and in microemulsions are different. Those in microemulsions are typically smaller, providing a greater solubilizing capacity for the system (PRINCE, 1975). This is of significant importance for D-limonene formulations, as it is desirable for the formulation to be capable of incorporating the highest possible concentration of the essential oil, given that its therapeutic dosages are typically high (25 to 300 mg/kg of body weight). Therefore, a higher content of D-limonene in the formulation allows for the administration of a smaller final product volume. This discussion was also incorporated into the manuscript (lines 313 - 327).

In light of the above, we reaffirm our choice to focus on microemulsion systems, despite the higher number of experiments required. When developing a new product, it is crucial to carefully consider the specific requirements of your application and the desired product properties. In this regard, we believe that microemulsions are better suited for this particular case. We greatly appreciate your understanding and your consideration of these factors.

  1. It is recommended to combine figures 2 to 13 into two or three figures with appropriate grouping based on similar characteristics or trends.

  Firstly, we would like to clarify that the figures were grouped according to the surfactant and HLB value used, providing the reader with an insight into the researchers' workbench. Besides that, we agree that an excess of images is unnecessary. As a result, we have retained only those where one or more microemulsions were obtained (Figure 2 and 3), while relocating the remaining images to the supplementary material.

  1. It is recommended to combine figures 17 to 23 into two or three figures with appropriate grouping based on similar characteristics or trends.

  The strategy applied to Figures 2 through 13 has also been implemented for figures 17 through 23, as described in the previous comment.

  1. It is strongly recommended to use the definition of micelle in describing section 3.2.2 Titration method and explain how it relates to the formation and stability of microemulsions.

  To address this request, the following discussion was included in lines 404 to 420: "It is worth highlighting, as demonstrated in Table 7, that while microemulsions commonly self-assemble and frequently require high concentrations of surfactants and co-surfactants to significantly reduce the interfacial tension between the oil and aqueous phases [21; 22], it was possible to formulate a product with desirable properties using only approximately 30% of a single surfactant. In this particular scenario, the bending elastic energy, typically of secondary importance in relation to surface energy, gains greater relevance within microemulsion systems due to their remarkably low interfacial tension [28].

This is because the bending elastic energy is directly related to the shape of the formed micelle, with the droplet shape being the most common [28]. Given that these are oil-in-water compositions, it is understood that the D-limonene droplets are surrounded by a surfactant layer and dispersed in water within the obtained microemulsions. The surfactant layer, in turn, has a maximum limit of expansion. When this limit is reached, it leads to the separation into various phase types, such as excess oil or water, presence of emulsions, among others. This phenomenon was likely observed during the titration D, at the point where the D20 formulation was unable to reach thermodynamic equilibrium.”

  1. In Figures 17 to 23, the formulation components are unclear. Please provide more explanation and refer to them in the figure captions.

  Such figures have been relocated to the supplementary material, resulting in the creation of new figures containing only microemulsion images. However, the captions in the supplementary material now include the starting mixture ratio for the titration, as exemplified below: “Figure S15. Formulation Obtained via Titration A, derived from a mixture of 55% D-limonene and 45% Labrasol”. We believe that this approach makes the composition of the formulations evident.

  1. Considering the molecular structure of Labrasol®, it seems that this molecule is prone to oxidation. How was accounted for in the oxidative stability test?

  We acknowledge that the presentation of the results may have raised questions regarding the oxidation of the formulation, and it may not have been entirely clear whether it pertained to D-limonene or other components. To address this concern, we have included a new graph (Figure 10) in the manuscript, and the following paragraph in Lines 510 - 517: “Figure 10 provides an example of the observed difference between the formulation and its corresponding placebo, indicating that Labrasol® oxidation in water starts only after the oxygen consumption of the microemulsion is completed. Thus, it is understood that oxidation observed in microemulsions is mostly, if not exclusively, due to D-limonene. Although it may not be possible to identify D-limonene byproducts such as limonene oxide, carvone, and perillyl alcohol through this technique [30], it can serve as an alternative and indirect means to assess both the oxidation and volatilization of D-limonene in the absence of more robust techniques like GC-MS.

  1. As the optimal formulations have nearly 50% surfactant, it is strongly recommended to discuss the correlation between micelle formation and stability in a separate section in the text.

Firstly, we would like to emphasize that formulations containing less than 50% of the surfactant were achieved. However, those with a higher concentration of D-limonene fell within the range of 45 to 55% Labrasol.

In order to provide a more comprehensive explanation of the correlation between the formed micelles and the oxidative stability of the formulations, the following discussion has been included in lines 561 to 572: "When evaluating the composition of microemulsions, it becomes evident that the concentration of Labrasol in these formulations varies very little, and does not exert a significant influence on the induction period, unlike the quantities of D-limonene and water. This phenomenon may be attributed to the application of a temperature of 120°C during the assay. As demonstrated further in the thermal stress study, phase inversion of the microemulsion occurs from 45°C onwards, potentially leading to the destabilization of formed micelles. In other words, the elevation of temperature within the assay chamber results in a higher solubility of Labrasol in the oil phase compared to the aqueous phase [32]. Within this context, a higher concentration of D-limonene in the formulation leads to a heightened susceptibility of micelle destabilization with temperature escalation, making D-limonene more accessible to oxidation, and consequently, reducing the induction period."

  1. The number of references used in this study is insufficient. It is recommended to review and cite relevant and valid studies in the discussion section to support the findings and compare them with the existing literature.

We apologize for the previously insufficient number of references. To address this, we have included the following references, which were utilized throughout the text to support the new discussions brought forth: 

  1. KARADAÄž, AyÅŸe Esra; OKUR, Neslihan ÜstündaÄŸ; DEMIRCI, Betül; DEMIRCI, Fatih. Rosmarinus officinalis L. Essential Oil Encapsulated in New Microemulsion Formulations for Enhanced Antimicrobial Activity. J. Surfactants Deterg. 2021, 25(1), 95-103. Wiley. http://dx.doi.org/10.1002/jsde.12549.
  2. TOLEDO, et al. Improved in vitro and in vivo Anti-Candida albicans Activity of Cymbopogon nardus Essential Oil by Its Incorporation into a Microemulsion System. Int. J. Nanomedicine 2020, 15, 10481-10497. Informa UK Limited. http://dx.doi.org/10.2147/ijn.s275258.
  3. LAOTHAWEERUNGSAWAT, Natnaree et al. Transdermal Delivery Enhancement of Carvacrol from Origanum vulgare L. Essential Oil by Microemulsion. Int. J. Pharm. 2020, 579, 119052. Elsevier BV. http://dx.doi.org/10.1016/j.ijpharm.2020.119052.
  4. MEHANNA, Mohammed M.; ABLA, Kawthar Khalil; ELMARADNY, Hoda A. Tailored Limonene-Based Nanosized Microemulsion: Formulation, Physicochemical Characterization and In-Vivo Skin Irritation Assessment. Adv. Pharm. Bull. 2020, 11(2), 274-285. Maad Rayan Publishing Company. http://dx.doi.org/10.34172/apb.2021.040.
  5. Herneisey, M. et al. Quality By Design Approach Utilizing Multiple Linear And Logistic Regression Modeling For Microemulsion Scale Up. Molecules, [S.L.], V. 24, N. 11, P. 2066, May 30, 2019. MDPI Ag. Http://Dx.Doi.Org/10.3390/Molecules24112066.
  6. AZAD, Sk.; MEERAVALI, Sk. N.; BABU, P. C.; KUMAR, K. R.; NAIK, V.V. MICRO EMULSIONS: an overview and pharmaceutical applications. World J. Curr. Med. Pharm. Res. 2020, 2(2), 201-205. http://dx.doi.org/10.37022/wjcmpr.2020.2222.
  7. PAVONI, L.; PERINELLI, D. R.; BONACUCINA, G.; CESPI, M.; PALMIERI, G. F. An Overview of Micro- and Nanoemulsions as Vehicles for Essential Oils: Formulation, Preparation and Stability. Nanomaterials 2020, 10(1), 135. MDPI AG. http://dx.doi.org/10.3390/nano10010135.
  8. PRINCE, Leon M. Microemulsions versus Micelles. J. Colloid Interface Sci. 1975, 52(1), 182-188. Elsevier BV. http://dx.doi.org/10.1016/0021-9797(75)90315-x.
  9. BIZZO, H. R.; HOVELL, A. M. C.; REZENDE, C. M. Óleos Essenciais no Brasil: Aspectos Gerais, Desenvolvimento e Perspectivas. Quím. Nova 2009, 32(3), 588-594. FapUNIFESP (SciELO). http://dx.doi.org/10.1590/s0100-40422009000300005.
  10. SUN, J. D-Limonene: Safety and Clinical Applications. Altern. Med. Rev. 2007, 12(3), 259-264.
  11. KOMIYA, M.; TAKEUCHI, T.; HARADA, E. Lemon Oil Vapor Causes an Anti-stress Effect via Modulating the 5-HT and DA Activities in Mice. Behav. Brain Res. 2006, 172(2), 240-249. Elsevier BV. http://dx.doi.org/10.1016/j.bbr.2006.05.006.
  12. SOULIMANI, R.; BOUAYED, J.; JOSHI, R. K. Limonene: Natural Monoterpene Volatile Compounds of Potential Therapeutic Interest. Am. J. Essent. Oil Nat. Prod. 2019, 4(7), 01-10.
  13. LANGEVIN, D. Micelles and Microemulsions. Annu. Rev. Phys. Chem. 1992, 43(1), 341-369. Annual Reviews. http://dx.doi.org/10.1146/annurev.pc.43.100192.002013.
  14. Kashid, S. et al. Elevated Yield of Fragrance Compounds: Biotransformation of d-Limonene via Whole-Cell Immobilization of Pseudomonas putida and Rhodococcus erythropolis. Journal of the Institution of Engineers (India): Series E, [S.L.], v. 104, n. 1, p. 83-93, October 31, 2022. Springer Science and Business Media LLC. http://dx.doi.org/10.1007/s40034-022-00252-6.
  15. Bendjaballah, M.; Canselier, J. P.; Oumeddour, R. Optimization of Oil-in-Water Emulsion Stability: experimental design, multiple light scattering, and acoustic attenuation spectroscopy. Journal Of Dispersion Science And Technology, 2010, 31(9), 1260 - 1272. http://dx.doi.org/10.1080/01932690903224888.
  16. Ruckenstein, E. Phase Inversion Temperatures of Macro- and Microemulsions. Langmuir, v. 13, p. 2494-2497, February 20, 1997.

Once again, we extend our gratitude for your valuable insights and feedback.

Kind regards.

Round 2

Reviewer 1 Report

Satisfied with the revision. 

Reviewer 3 Report

Dear Authors, 

After reading the answers carefully and checking them in the text, I found them sufficient and convincing. The responses demonstrate scientific rigor, logical coherence, and persuasive evidence. The authors are appreciated for their precision and diligence.

Regards,